

# Natural language processing with transformers: a review

Georgiana Tucudean[1], Marian Bucos[1], Bogdan Dragulescu[1] and Catalin Daniel Caleanu[2]

[1] Communications Department, Politehnica University Timişoara, Timişoara, Timiş, România
[2] Applied Electronics Department, Politehnica University Timişoara, Timişoara, Timiş, România

## ABSTRACT

Natural language processing (NLP) tasks can be addressed with several deep learning architectures, and many different approaches have proven to be efficient. This study aims to briefly summarize the use cases for NLP tasks along with the main architectures. This research presents transformer-based solutions for NLP tasks such as Bidirectional Encoder Representations from Transformers (BERT), and Generative Pre-Training (GPT) architectures. To achieve that, we conducted a step-by-step process in the review strategy: identify the recent studies that include Transformers, apply filters to extract the most consistent studies, identify and define inclusion and exclusion criteria, assess the strategy proposed in each study, and finally discuss the methods and architectures presented in the resulting articles. These steps facilitated the systematic summarization and comparative analysis of NLP applications based on Transformer architectures. The primary focus is the current state of the NLP domain, particularly regarding its applications, language models, and data set types. The results provide insights into the challenges encountered in this research domain.

## INTRODUCTION

As natural language processing (NLP) tasks become more extensive, the processes involved in understanding human language become a challenge to keep up with. NLP encompasses a vast level of computational processes combined with linguistic fundamentals, subtly composing the Artificial Intelligence (AI) subfield that studies the relationship between computer and human language understanding.

The NLP domain has gained interest because of its broad applicability. Over time, it has successfully addressed different types of problems, such as information extraction, sentiment analysis, text summarization, information exchange, speech enhancement, translation, part of speech (POS) tagging, named entity recognition (NER), text classification, content generation, or even other complex approaches in the medical or educational fields. NLP tasks can be approached using several concepts, such as Transformers (*Acheampong, Nunoo-Mensah & Chen, 2021*; *Gao et al., 2021*; *Lukovnikov, Fischer & Lehmann, 2019*; *Le et al., 2021*), neural networks (*Le et al., 2021*; *Al-Yahya et al., 2021*), deep learning (*Fu, 2019*; *Colón-Ruiz & Segura-Bedmar, 2020*; *Xie et al., 2021*). Even

Corresponding author
Georgiana Tucudean,
georgiana.tucudean@upt.ro

considering its versatility, continuous expansion, and improvement, the field of NLP had reached some limitations. Some of the general problems that occur while addressing an NLP task are issues related to limitations of language concepts (*Yang et al., 2020*; *Gidiotis & Tsoumakas, 2020*; *Zhang et al., 2019*). Another issue is related to the characteristics of speech and specific language expressions used in conversations that cannot be identified automatically or fully understood by NLP methods (*Mozafari, Farahbakhsh & Crespi, 2020*; *Sohn & Lee, 2019*).

NLP is a subfield of AI that allows computers to understand and generate human language. It consists of two parts: natural language understanding (NLU) and natural language generation (NLG). The first part mentioned above, NLU, represents all the concepts involved in the process of understanding natural language by computers. NLU allows the computer to understand the context information for different forms of data and languages. The second part, NLG, refers to the process of data generation—phrases, sentences, and paragraphs—based on an internal representation. NLG gives meaning to phrases and sentences by following standard steps: identification of goals, planning to achieve goals based on evaluation and available sources, and finally, incorporation of plans into text (*Khurana et al., 2023*).

Some of the most popular approaches that have recently proven their efficiency in context of NLP tasks, and to which we will refer in this study, are the pre-trained models, Transformers, such as Bidirectional Encoder Representations from Transformers (BERT) and Generative Pre-Training (GPT). On an architectural level, BERT was primarily designed for NLU tasks, more specifically, for encoding text representations, while GPT-2 was developed for language modeling purposes, as a decoder-only architecture (*Rothe, Narayan & Severyn, 2020*). The standard workflow of BERT models consists of two tasks: pre-training and fine-tuning. The last step allows the pre-trained model to be fine-tuned for specific tasks such as: question answering, summarization, translation, *etc.* BERT models incorporate multiple levels of language knowledge, such as syntactic knowledge, semantic knowledge, and world knowledge (*Rogers, Kovaleva & Rumshisky, 2020*).

Although both the BERT and GPT models have demonstrated good performance in NLP tasks, their architectures present fundamental differences. BERT has an encoder-only transformer architecture. It relies on a bidirectional Transformer architecture which allows the model to capture the context from both proximities of the words. GPT is based on the decoder-only transformer; the model uses a unidirectional transformer architecture, processing text in a manner closer to the most common human reading/writing direction. This empowers GPT to perform text predictions, but the model is limited in understanding the full context around a given word. These specific characteristics make the two models more suitable for different scenarios: BERT models are suitable for search or classification problems, while GPT models are efficient for text generation tasks.

Over the years, GPT has emerged on an increasing scale of improvements. GPT-1 uses a 12-layer decoder-only transformer with masked self-attention. GPT-1 is a language pre-trained on 7,000 unpublished books and consists of 117 million parameters. GPT-2 improves GPT-1 with a few changes, the network consists of a 48-layer Transformer with 1.5 billion parameters, an increase in context size from 512 to 1,024 tokens, and introduces

the concept of task conditioning that allows learning multiple tasks using the same unsupervised model (*Lauriola, Lavelli & Aiolli, 2022*; *Radford & Narasimhan, 2018*). Furthermore, GPT-2 is trained using larger and more diverse data sets, this makes it powerful and capable of addressing a wide range of language tasks. The new GPT-3 is an autoregressive model that outperforms GPT-2, and its enhanced ability to generate new data based on past values contributed to improved results in text generation tasks. GPT-3 has 175 billion parameters and 96 decoder layers and has proven better efficiency in the text generation process (*Lauriola, Lavelli & Aiolli, 2022*). GPT-3 is trained on a 500 billion word data set; this makes the model faster and more powerful. The model eliminates the necessity of fine-tuning and has been shown to generate text data that are very similar to human-generated text. Along with its impressive performance, GPT-3 raises significant concerns about its ethical and social implications, leaving room for future studies to explore.

## Rationale

Regarding related work, *Lin et al. (2022)* conducted a comprehensive review of Transformers. The authors explored different architectures of Transformers, highlighting their strengths and weaknesses. They also discussed modifications made to the original Transformer architecture, such as the introduction of self-attention mechanisms and positional encodings. Furthermore, *Lin et al. (2022)* proposed a taxonomy to categorize different types of Transformers based on their architectural characteristics. This taxonomy serves as a useful framework for researchers and practitioners to better understand the design choices and trade-offs involved in implementing Transformers. Although the paper by *Lin et al. (2022)* covered a wide range of applications of Transformers, it did not specifically focus on their application in NLP tasks.

In comparison with the existing studies, in this review we present a structured overview of NLP by addressing the domain of applicability, the problems that NLP solves, the existent architectures, and the data sources that can be used. To achieve the above-mentioned goals, we want to address the following research questions: i) What is the current status of the NLP Transformers concerning its applications, language models, and data sets? ii) What are the limitations and challenges of NLP Transformers, and how have researchers attempted to address them?

To summarize, this review contributes with a structured assessment of the challenges and solutions that NLP transformer-based approaches encompass, along with the problems that can occur: language concepts limitations, specific language characteristics, and expressions. Furthermore, we look at some of the methods that have been implemented to improve the performance of NLP tasks (*Xie et al., 2021*; *Rothe, Narayan & Severyn, 2020*; *Ham et al., 2021*). Taking into account the wide topics that NLP addresses and the solutions that continue to evolve, this study aims to provide an overview of the domains that can be addressed with NLP and the new approaches that occur in this field of study.

## Intended audiences

This review is intended for two groups of audiences who share a common ground. The first group is represented by linguistic specialists who are familiar with the domain of

applicability for NLP but aim to identify the new transformer-based architecture solutions in this field of study. And the second group includes computer science developers who are familiar with the technical implementations but want to understand what are the problems that can be addressed in NLP domain, considering its broad solutions: classification tasks, translation, text summarization, text augmentation, sentiment analysis, *etc*. The first goal is to emphasize the promising results that transformer-based architectures provide in the context of NLP tasks and encourage the linguistic specialists to experiment with these approaches with various text data sets. NLP with transformers is more popular in the context of common languages such as English, and our aim is to encourage the expansion of these models in various languages. The second goal is to serve as an inspiration for data scientists to address NLP tasks for different problems.

The structure of this review is organized as follows: in the next section, we present the survey methodology we followed to perform this review for NLP with Transformers. The survey includes Protocol development, Inclusion and Exclusion criteria, Quality assessment and Data extraction process. The Results section presents a systematic review of the selected articles based on the domain of applicability for NLP, common data sources, and architecture approaches, followed by Discussions, and Conclusions, to emphasize the problems that NLP can solve and to summarize the outcomes of this study.

## SURVEY METHODOLOGY

For this review, we refer to the method proposed by *Petersen, Vakkalanka & Kuzniarz (2015)*. The objective was to identify and extract a set of articles that are relevant to our topic. After that, we had to adapt our guidelines to acquire the most important studies to refer to in the review. A more detailed description of the article selection algorithm is presented in the following steps.

### Protocol development

The extraction process was performed on the following platforms: Scopus, Web of Science, and IEEE. We consider a 5-year period, from 2018 to 2022. The search method was based on the following key terms: "natural language processing", "NLP", "transformer*". Key terms for title, abstract, and keywords were included. At the end of this step, we obtained 3,387 unique items from a total set of 5,321 items.

### Inclusion and exclusion criteria

To be able to conduct the review, we had to filter the articles to reduce the number of studies that are taken into consideration for the next step. Our objective was to include the most relevant articles. The selection process is further detailed in the following steps:

   1. Exclusion criteria
- Language: exclusive articles in English.
- Completion availability: full-text articles.
- Review Process: peer-reviewed articles.

2. Inclusion criteria
- Category: Computer Science related articles.
- Raw data: BibTeX availability.
- Good publishers: articles in journals and conferences.
- Total Smart Citations: provides a granular perspective on the impact of individual references within the text, without grouping them at the citation publication level (*Nicholson et al., 2021*; *Bakker, Theis-Mahon & Brown, 2023*). By using Total Smart Citations, we aim to ensure a comprehensive assessment of the relevance of each reference cited within the articles we considered in the review. Specifically, by setting a threshold of Total Smart Citations rank greater than 24, we aimed to focus our attention on articles that have a high level of influence across the academic community, thus enhancing the credibility and robustness of our selection methods.
- Relevance: emphasize the domains addressed in the current study.

After the inclusion and exclusion process, we obtained a set of 128 relevant studies to consider in the next phase.

## Quality assessment

As a first point, we analyze the title and abstract section of each study. This was the preliminary step of the quality assessment process and the goal was to identify the main topics that apply Transformers for NLP tasks.

As a second point, we conducted an in-depth analysis of the research methodology and the theoretical framework presented in the articles. We evaluated the relevance of the subject for each study, with particular attention to the methodology and related work section. If one of these sections was missing, the article was excluded for the next step because the lack of this specific information would affect in a negative way the understanding and comparison of the NLP models and architectures. After this phase, we obtained 42 articles to consider for the data extraction process.

## Data extraction

At this point, we conducted a closer analysis of each selected article. The goal was to extract the following information: keywords, the study domain (*i.e.*, topic), the objective, the methodology, the tools used in the experiment, the source of the input data, the results, and finally the limitations that occurred in the study.

The overall methodology that we approach in this review is depicted in Fig. 1. The search and selection process of the research platforms concluded with 3,387 unique studies from the initial set of 5,321 items. Inclusion and exclusion criteria narrowed the quality assessment process to 128 studies. After assessing the relevance of the subject and the overall workflow, we obtained 42 items for the data extraction process.

Based on the information extracted, we decided to exclude studies that do not include a detailed experimental strategy. Studies that describe the experimental protocol helped us facilitate the comparative study. Understanding the NLP processes and architectures

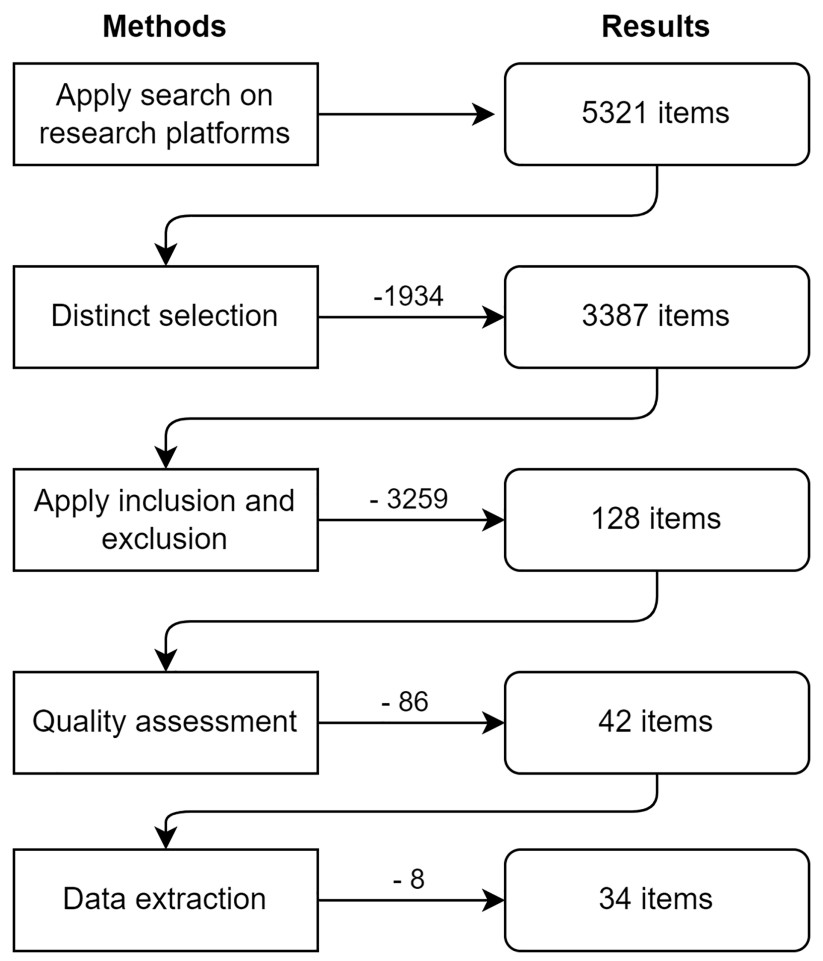

**Figure 1 Steps applied for extraction methodology.**

helped to conduct a structured manner for the review process. In the end, the data extraction process ended with 34 of the most relevant items to consider for further review.

## RESULTS

The selected articles present NLP methods using Transformers or hybrid models, which currently represent the widespread standard approach in practice. We managed to label the articles according to the problems addressed.

We conclude with the following labels: classification, optimization, content generation, sentiment analysis, text summarization, and Named Entity Recognition (NER). The results are presented in Tabel 1.

1. Text classification

This topic includes studies that aim to predict the label of input data. The selected articles address multiclass classification and binary classification.

2. Optimization

This label includes articles that present an approach to improve an existing model, either at an architectural level (*e.g.*, (*Xie et al., 2021*; *Rothe, Narayan & Severyn, 2020*)) or at a hardware level (*e.g.*, (*Ham et al., 2021*; *Zafrir et al., 2019*)).

**Table 1 Distribution of articles topic.**

| Topic name | Number of articles | Articles |
|---|---|---|
| Text classification | 12 | Le et al. (2021), Farahani et al. (2021), Fan et al. (2020), Whang et al. (2020), Ayoub, Yang & Zhou (2021), Radfar, Mouchtaris & Kunzmann (2020), Yang et al. (2020), Yu, Su & Luo (2019), Sung, Dhamecha & Mukhi (2019), Rasmy et al. (2021), Chang et al. (2020), Balagopalan et al. (2020) |
| Content generation | 6 | Nguyen et al. (2019), Li et al. (2021), Liu et al. (2020), Mastropaolo et al. (2021), Sharma et al. (2022), Bagal et al. (2022) |
| Sentiment analysis | 5 | Mozafari, Farahbakhsh & Crespi (2020); Sohn & Lee (2019); He et al. (2021), Potamias, Siolas & Stafylopatis (2020), Zhang et al. (2020) |
| Text summarization | 4 | Gidiotis & Tsoumakas (2020), Yang et al. (2020), Lee et al. (2020), Gavrilov, Kalaidin & Malykh (2019) |
| Optimization | 4 | Xie et al. (2021), Rothe, Narayan & Severyn (2020), Ham et al. (2021), Zafrir et al. (2019) |
| NER | 3 | Yang et al. (2020), Zhang et al. (2019), Souza, Nogueira & Lotufo (2020) |

3. Content generation

This topic refers to studies that aim to empower the meaning of data.

4. Sentiment analysis

The topic refers to articles that aim to identify people's opinions based on input text data.

5. Text summarization

This topic includes studies that apply NLP processes to reduce long text into shorter paragraphs by removing some of the less relevant information.

6. Named entity recognition

This label includes articles that address the well-known NLP challenge, named entity recognition. The goal of NER tasks is to identify entities in text data and further classify names into specific domains.

The topic labeling was an important process for the current research because one of the objectives was to identify the primary applications of NLP. Table 1 presents the distribution of articles according to the topics addressed.

Based on the quality assessment process and the information presented in Table 1, it can be observed that the most common problem addressed by the NLP topic is the classification problem. Text classification is a common problem in this research area because text information is defined by unstructured data types. Since text data tends to be erratic, most organizations struggle to manage large volumes of unorganized text data, leading to inefficient utilization of valuable information. NLP classification tasks successfully overcome this common problem.

Another interesting feature relevant to this review is the data sources considered in the selected studies.

Table 2 presents the data sources used in the articles we consider in this review. We were able to classify the types of data sets as follows: (1) specialized data sets—specific data that can be used for a targeted domain and/or data collected in collaboration with an organization; (2) publicly available data sets—data that are open source and can be easily accessed; (3) benchmark data sets—data sets are compiled to develop, test, and compare

**Table 2 Distribution of data source types.**

| Data set type | Number of articles | Articles |
|---|---|---|
| Specialized | 16 | *Xie et al. (2021)*, *Zhang et al. (2019)*, *Mozafari, Farahbakhsh & Crespi (2020)*, *Rothe, Narayan & Severyn (2020)*, *Ham et al. (2021)*, *Whang et al. (2020)*, *Yang et al. (2020)*, *Sung, Dhamecha & Mukhi (2019)*, *Rasmy et al. (2021)*, *Chang et al. (2020)*, *He et al. (2021)*, *Potamias, Siolas & Stafylopatis (2020)*, *Zhang et al. (2020)*, *Yang et al. (2020)*, *Lee et al. (2020)*, *Souza, Nogueira & Lotufo (2020)* |
| Publicly available | 9 | *Yang et al. (2020)*, *Gidiotis & Tsoumakas (2020)*, *Fan et al. (2020)*, *Ayoub, Yang & Zhou (2021)*, *Radfar, Mouchtaris & Kunzmann (2020)*, *Balagopalan et al. (2020)*, *Liu et al. (2020)*, *Sharma et al. (2022)*, *Gavrilov, Kalaidin & Malykh (2019)* |
| Benchmark | 6 | *Le et al. (2021)*, *Farahani et al. (2021)*, *Nguyen et al. (2019)*, *Yu, Su & Luo (2019)*, *Mastropaolo et al. (2021)*, *Bagal et al. (2022)* |
| Created | 3 | *Sohn & Lee (2019)*, *Zafrir et al. (2019)*, *Li et al. (2021)* |

the performance of different algorithms to identify the most effective solution for a given problem (*Dhar & Shamir, 2021*); (4) created data sets—studies in which the data sets were created as a personal contribution with specific qualities to address particular tasks. As shown in Table 2, many studies leverage publicly available data sets, within domains such as biomedicine. For example, the clinical study for concept extraction using Transformers (*Yang et al., 2020*) uses multiple specialized data sets from the biomedical domain, an example is the biomedical data set for relation classification and entity typing (paperswithcode.com/dataset/2010-i2b2-va). Other studies, such as hate speech detection in online social networks (*Mozafari, Farahbakhsh & Crespi, 2020*), rely on specialized data sources customized to address specific problems, for example, annotated tweets data sets for hate speech detection (GitHub.com/zeeraktalat/hatespeech). Studies such as Persian language understanding using Transformer models (*Farahani et al., 2021*) utilize benchmark data sets dedicated to academic research purposes: manually annotated named-entity data sets in Persian language (GitHub.com/HaniehP/PersianNER). Additionally, we have identified studies focused on data set creation (*Li et al., 2021*) that use sentence generation for Audio-Visual Dialog data set (GitHub.com/dialogtekgeek/DSTC8-AVSD_official).

Taking into account the various topics addressed within the NLP domain, it can be observed that the majority of data sets are specialized data sets (Table 2). This is an expected outline considering the versatility of NLP methods and the good performances that DL algorithms have demonstrated over the last few years.

On an architecture technology level, the studies mainly used transformer-based approaches, but hybrid approaches that combine different types of neural networks are also utilized-BERT and NN, BERT and GPT2, and Transformers and NN, sections referred to in Table 3. The most popular architecture identified was BERT. Some articles present interesting approaches that include GPT/GPT2 architectures. We can also mention other Transformers such as the Evolved Transformer (*Nguyen et al., 2019*), the X-Transformer (*Chang et al., 2020*), the T5 (*Mastropaolo et al., 2021*), the Universal Transformer (*Gavrilov, Kalaidin & Malykh, 2019*), *etc.* An overview of the architectures is shown in

**Table 3 Distribution of language models.**

| Architecture type | Number of articles | Articles |
|---|---|---|
| BERT | 15 | *Yang et al. (2020)*, *Zhang et al. (2019)*, *Mozafari, Farahbakhsh & Crespi (2020)*, *Sohn & Lee (2019)*, *Farahani et al. (2021)*, *Zafrir et al. (2019)*, *Radfar, Mouchtaris & Kunzmann (2020)*, *Yang et al. (2020)*, *Sung, Dhamecha & Mukhi (2019)*, *Rasmy et al. (2021)*, *Balagopalan et al. (2020)*, *Potamias, Siolas & Stafylopatis (2020)*, *Yang et al. (2020)*, *Lee et al. (2020)*, *Souza, Nogueira & Lotufo (2020)* |
| GPT/GPT2 | 4 | *Li et al. (2021)*, *Liu et al. (2020)*, *Sharma et al. (2022)*, *Bagal et al. (2022)* |
| BERT and NN | 2 | *Le et al. (2021)*, *Mastropaolo et al. (2021)* |
| BERT and GPT2 | 2 | *Rothe, Narayan & Severyn (2020)*, *Chang et al. (2020)* |
| Other transformers | 11 | *Xie et al. (2021)*, *Gidiotis & Tsoumakas (2020)*, *Ham et al. (2021)*, *Fan et al. (2020)*, *Whang et al. (2020)*, *Ayoub, Yang & Zhou (2021)*, *Nguyen et al. (2019)*, *Yu, Su & Luo (2019)*, *He et al. (2021)*, *Zhang et al. (2020)*, *Gavrilov, Kalaidin & Malykh (2019)* |

Table 3. The following subsections provide a detailed analysis of the methods and architectures. The analysis is structured according to the identified categories of Transformers. Initially, we discuss the BERT-based approaches and hybrid methods, followed by GPT, and concluding with Other Transformers.

## Bert approaches and hybrid methods

Based on the results presented in Table 3 and the overall analysis, the NLP BERT-based approaches outperform the other methods when considering any of the identified problems: text classification or summarization, content generation, sentiment analysis, NER, or optimization tasks. The BERT architecture, introduced in 2018 by Google (*Devlin et al., 2018*), leverages both left and right contexts in all layers. As seen in Table 3, the most popular architectures used in the selected studies are BERT-based approaches or hybrid approaches that include BERT. For example, in the context of clinical concept extraction, *Yang et al. (2020)* explored four widely used transformer-based architectures, BERT, RoBERTa, ALBERT, and ELECTRA. The aim was to extract various types of clinical concepts from three public data sets. In the pretraining process, the authors experimented with general transformer models using general English *corpus* and clinical transformer models pre-trained with clinical *corpus*. In the fine-tuning stage, they added a linear classification layer to predict named entities using labeled clinical concepts from the training set. The parameters of the transformer models and the parameters of the classification layer were optimized to extract of clinical concepts. The results were compared with a Long Short-Term Memory Conditional Random Fields (LSTM-CRFs) model as a baseline. RoBERTa outperformed the baseline results; the best results for the F1 score were around 0.8 for the three data sets, but ALBERT and ELECTRA achieved comparable results. However, this study has an important limitation, it is mainly focused on clinical concept extraction, a word-level task for NLP. In contrast, other recent studies have explored BERT for sentence level and have obtained promising results for applications such as semantic textual similarity, clinical records classification, or question-

answering (*Yang et al., 2020*). Also, for the NER topic, an interesting study proposed by *Souza, Nogueira & Lotufo (2020)* aims to train BERT models for Brazilian Portuguese. The addressed NLP tasks are sentencing textual similarity, recognizing textual entailment, and named entity recognition. The BERT-based architecture model was trained using different layer sizes (Base-12 layers and Large-24 layers) and managed to improve the state-of-the-art for all the proposed tasks. As future research, the study presented in this article (*Souza, Nogueira & Lotufo, 2020*) proposes to experiment with other new and efficient models such as RoBERTa and T5. In the area of sentiment analysis, *Mozafari, Farahbakhsh & Crespi (2020)* introduced a BERT-based transfer learning approach to identify hateful speech within online social media content. Their goal was to train a classifier with different layers on top of the pre-trained $BERT_{base}$ transformer to minimize task-specific parameters. Given that the BERT model is pre-trained on general *corpus*—English Wikipedia and Book *Corpus*, their strategy was to analyze contextual information extracted from the pre-trained layers of BERT and fine-tune the information using annotated data sets. They managed to outperform previous works on tweet classification for all metrics considered (F1 score, Waseem, and Davidson). The study identifies a common challenge: many errors occur due to biases in data collection and annotation rules. The problem of detecting hate speech is challenging and has raised some difficulties due to the nature of the language. False positive classifications of hate speech can constrain the freedom of expression of online users, while false negative classifications of hate speech can negatively impact the overall well-being of online communities (*Mozafari, Farahbakhsh & Crespi, 2020*). However, conducting a comprehensive examination of contextual information embedded within BERT's layers, combined with analysis of various features associated with different types of biases, facilitated the detection and mitigation of biased data. This represents an interesting contribution of the study, as it offers a possible solution to one of the common challenges within the hate speech detection issue. *Sohn & Lee (2019)* take a similar approach to the topic of sentiment analysis. This article proposes a BERT-based multichannel model for hate speech detection in multiple languages. The strategy for this implementation consists of data set collection and processing, using translation to create data in other languages, fine-tuning BERT for sentence classification, and creating multichannel BERT architecture for the considered languages. Like *Mozafari, Farahbakhsh & Crespi (2020)*, in the fine-tuning approach, the authors focused on setting the $BERT_{base}$ parameters, along with a Softmax operation to normalize the output and obtain values between 1 and 0. The proposed strategy obtained good results for all data sets in terms of F1 score and accuracy. For the problem addressed, the study encounters some challenges for automatic detection of hate speech: first, the characteristics of the language are very different from one country to another, and second, the nature of the language (sarcasm/swear words) can cause some difficulties for the classifier's ability to understand multiple characteristics. A common problem for sentiment analysis tasks is to accurately identify intentions, given the various ways sentiments like irony, hate, sarcasm, *etc.*, can be expressed. An important finding of this study is that despite introducing errors in translation, this process adds additional value to the input, improving the classification

results. Additionally, this study applies transfer learning to overcome the challenge of small data sets, which represents a well-known solution for this NLP problem.

As a solution for language problems that can occur in the context of sentiment analysis, fine-tuning BERT on massive sets of data can be effective; this enables transformers to incorporate multiple language characteristics and overcome challenges similar to those identified in the previous studies.

For the classification field, *Yang et al. (2020)* designed a new hierarchical transformer using Whole Word Masking BERT, with a multitasking architecture that uses text and audio data from quarterly earnings conference calls and predicts future price volatility in the short and long term. The proposed HTML model contains four components: a token-level transformer encoder, multimedia information fusion, a sentence-level transformer encoder, and multitask prediction. First, text and audio features are extracted from raw text/audio call content. Text tokens are derived from the text data and encoded into vectors using a pre-trained language model, while 27 audio features are extracted from audio data. The extracted text and audio features are combined in the information fusion layer and utilized as input to the sentence-level transformer encoder, which generates a new intermediate multimodal representation. This representation serves as input for the multitask learner. Finally, the multitask prediction layer generates predictions based on inputs from the sentence-level transformer encoder. The proposed method demonstrates good prediction accuracy, in the range of 17% to 49%, compared to other state-of-the-art methods. BERT-based architectures have successfully addressed classification problems in the medical field. For this topic, *Rasmy et al. (2021)* propose Med-BERT, a disease prediction model that uses electronic health records. The model is pre-trained on a gigantic and structured electronic health records data set and fine-tuned on a validation set that was pre-established. The model managed to gain impressive accuracy and AUC values for the disease prediction problem. Despite the results obtained, *Rasmy et al. (2021)* mentioned some of the limitations that occurred for the proposed model: the format of the input information is limited, and some parameters were omitted (time intervals between visits), this can lead to loss of temporal information. Another point is that in the experiment, the authors did not leverage the medical order for each medical visit, which can lead to important information loss. These observations outline the importance of leveraging all available information from input data when training and fine-tuning a transformer-based model. Furthermore, the preprocessing step plays a major role in defining input characteristics. A thorough analysis of the input data is crucial to identify key attributes, ultimately having a major contribution to the performance of the language models. Well-structured data and multiple features can lead to better results in terms of prediction problems. In the medical domain, the study (*Balagopalan et al., 2020*) uses an NLP BERT-based method to detect a predisposition to Alzheimer's disease. It shows that fine-tuning BERT models can perform well for Alzheimer's disease detection and outperform handcrafted feature engineering approaches. A thorough refinement in the strategy of this study is the transfer learning approach: to leverage the language information encoded by BERT, the authors used pre-trained model weights to initialize the model. The fine-tuning is done on training data using 10-fold cross-validation, and the

learning rate was improved using stochastic optimization and linear scheduling. The proposed BERT model performed well and uses as an evaluation the well-known metrics F1, accuracy, precision, recall, and specificity. On the limitations side, it can be mentioned that the accuracy for BERT-based models is high but very close to the values obtained for the classic ML approach, SVM (Support Vector Machine). For future research, they proposed a fusion model that combines BERT with ML methods. Hybrid methods can increase the performance of this type of detection task that combines the linguistic and acoustic features of speech (*Balagopalan et al., 2020*). We identified efficient BERT-based approaches to various tasks, in addition to those mentioned earlier. Some of the tasks are text summarization or optimization. An example of the optimization topic is reflected in the study proposed by *Zafrir et al. (2019)*. The purpose of this study is to reduce the number of resources used by transformers, such as computational, memory, and power resources. The strategy was to perform quantization-aware training while fine-tuning the phase of BERT to compress it four times with minimal accuracy loss; furthermore, the produced quantized model can accelerate the inference speed if it is optimized for 8-bit integer supporting hardware (*Zafrir et al., 2019*). The method was shown to be efficient and can enable low latency in NLP applications on various hardware platforms, from edge devices to data centers.

## GPT approaches

In this systematic review, we studied multiple approaches and identified state-of-the-art deep neural network methods for NLP. Some of the most promising methods are the GPT/GPT-2 models. The GPT architecture proved efficiency in different cases and managed to obtain impressive performance for topics such as content generation (*Li et al., 2021*; *Liu et al., 2020*; *Sharma et al., 2022*), or optimization strategies for various tasks: Machine Translation, Text Summarization, Sentence Splitting, and Sentence Fusion (*Rothe, Narayan & Severyn, 2020*). On the same topic of GPT, we identified molecular generation studies that aim to control the properties of multiple molecules (*Bagal et al., 2022*).

For content generation tasks, *Li et al. (2021)* proposed a universal multimodal transformer based on the GPT2 architecture to combine visual and textual representations and capture the interaction between different multimodal information—video, audio, video caption, and dialog context, understand dialogs, and generate informative responses. They used a special data set called the Audio-Visual Scene-Aware Dialog (AVSD) from DSTC7 and DSTC8. *Li et al. (2021)* propose a multitask learning method to learn representations among different types of information by fine-tuning language models. The aim is to capture information across both visual and textual data. To pursue their objective, the authors created a universal multimodal transformer that relies on fine-tuning processes. Fine-tuning processes include three tasks: response language modeling conditioned on video, audio, caption, and dialogue history; video-audio sequence modeling conditioned on caption and dialogue; caption language modeling conditioned on video and audio. The first task, response Language Modeling aims to generate responses based on video-audio features, captures, dialog history and questions by minimizing the log-likelihood loss function. The second task, video-audio sequence modeling, predicts

video-audio features, given caption, and dialog history using the video-audio feature regression method. Specifically, the second task regresses the Transformer output of the video-audio feature to the next video-audio feature. The third task, caption language modeling, trains the model to generate captions based on the video-audio feature by minimizing the negative log-likelihood loss function. The proposed model successfully learns representations across different types of information and generates informative responses. *Li et al. (2021)* conducted an objective evaluation for both data sets and obtained an impressive 98.4% of human performance based on human ratings. As future improvements, *Li et al. (2021)* plan to consider more video features for their experiment and explore different training tasks to improve the joint understanding of video and text. Another interesting approach to the topic of content generation presents code improvement tasks. The study presented by *Liu et al. (2020)* focuses on developing a pre-trained language model for multitask learning, specifically for code understanding and code generation. The experiment was carried out on open source real-world data sets for JavaScript and TypeScript programming languages. In the experimental setup, *Liu et al. (2020)* used a Transformer with six layers, 516 dimensional hidden states, six attention heads, and one inner hidden feed-forward layer. The model was pre-trained with a batch of 16 sequences for 600,000 steps. To assess their approach, they compared the model with state-of-the-art models, including neural network-based models—LSTM, and self-attentional neural network-based for code completion—Transformer-XL. The experimental results showed that the model outperforms previous state-of-the-art models and successfully adopts multitask learning for code completion. Their proposed model outperforms LSTM for both small and large test data sets, and the performance was substantially higher compared to Transformer-XL. Finally, in the field of GPT, another content generation approach addresses the problem of improving empathy in online mental health support with a deep reinforcement learning agent called Partner. The model is based on a transformer language model adapted from GPT-2 (*Sharma et al., 2022*). The goal is to transform low-empathy conversational posts into higher empathy and, at the same time, to maintain the quality of the conversation. The model handles two tasks: generating emphatic sentences and integrating them into appropriate textual contexts. This experiment relies on a very important resource: a specialized data set that serves as a solid foundation for building an empathic language model. To create a data set for empathic sentence generation, the authors used the largest peer-to-peer platform for mental health support. The data set consists of conversations between people seeking support and people who provide support. An important goal in this study was to analyze and filter out posts related to mental health. To do so, *Sharma et al. (2022)* manually annotated ~3k posts and trained a standard text classifier based on BERT, achieving an accuracy of ~85%. The classifier obtained was applied to the entire specialized data set, resulting in 3.33 M interactions from 1.48 M posts made by people seeking support. The strategy of acquiring a labeled data set from initially non-curated information using a BERT-based transformer is the novel aspect introduced by this study. For the empathic rewiring task and the other associated goals, *Sharma et al. (2022)* trained a reinforcement learning agent that learns when to stop making changes based on a special "stopping"

action. To do that, they had to take into consideration various aspects: the theoretical ground of empathy, the specificity of the context and the diversity of the response, text fluency, and sentence coherence, feedback rewriting, and training. The solution is based on the standard reinforcement learning framework consisting of a collection of states, a set of actions, a policy, and rewards. The principle is as follows: given a state, an agent takes an action based on a policy which dictates whether the agent should act in that state. The goal of the reinforcement learning agent is to learn a policy that maximizes the reward. The reinforcement learning model is designed to take advantage of the context from seeker posts to make empathic adjustments while operating on the response posts to identify areas for improvement. These adjustments are made in an adaptive manner, with a focus on ensuring minimal and precise changes through a dedicated "stopping" action. The reinforcement learning model constructs states based on seeker posts and fixed-length contiguous spans in response posts, defining actions as insertion, replacement, or deletion of sentences. The policy has a transformer language model based on GPT-2 that consists of a stack of masked multi-head self-attention layers. It takes as input an encoded representation of the state and generates an action. Overall, the policy is guided by a reward function that prioritizes empathic and flexible transformations while ensuring fluency, coherence, specificity, and diversity. To evaluate the results, the authors compared their approach with baseline approaches like DialoGPT, MIME, and BART. The Partner agent manages to demonstrate greater empathy and outperforms other baseline approaches. The Partner achieves the largest increase in empathy, 35% more than the next best approach, MIME. This GPT-2 based agent is an opportunity that contributes to the development and improvement of online conversational platforms. However, along with powerful solutions come additional responsibilities. To perform a thorough evaluation, human intervention was essential. Evaluating language generation is a challenge; therefore, for the previous study, six graduate students specializing in clinical psychology with expertise in empathy and mental health support were engaged. They evaluated the outputs generated by the partner model in comparison to those generated by other baseline models. Given these points, human input is indispensable to ensure quality and specificity in certain NLP tasks. To the best of our knowledge, human input has not yet been fulfilled by any form of artificial intelligence, even within the context of powerful models such as GPT.

## Other transformer approaches

As presented in Table 3, this review incorporates other transformer approaches known for their efficiency. *Nguyen et al. (2019)* present another study where the Transformer-based approach outperforms existing models, we have classified this study within the field of content generation. They proposed a transformer method to restore punctuation and capitalization for automatic speech recognition transcription that outperforms existing methods in both accuracy and decoding speed. An interesting point in this experiment is the preprocessing method. After cleaning up the characters by keeping only alphabet characters along with commas, stop words, and question marks, the authors had to make

sure that the punctuation was linked with the previous words to avoid syntactic ambiguity. Since the study addresses automatic speech recognition punctuation, the strategy relies on the chunking process for long inputs. The chunking component raises an interesting challenge: inaccurate predictions near the boundary of the chunk, due to insufficient left- and right-context information in that area. To overcome this challenge, the authors propose an overlapping strategy for consecutive chunks: long inputs are split into fixed-sized chunks (k words) with a sliding window of k/2. In the next phase, the real difficulty occurs when merging the overlapped results: because the output of the overlapped region between two consecutive chunks may be different, it is important to identify which words should be retained and which should be removed to form a complete sentence. This problem was mitigated by defining a parameter that allows flexible control over the words that overlap between consecutive chunks. This solution provides options to prioritize words from the first chunks or the second chunk based on the parameter value. To evaluate the impact of this parameter, *Nguyen et al. (2019)* experimented with the sequence-to-sequence LSTM model. The outcome shows that combining the chunk-merge strategy with the Evolved Transformer outperforms existing methods and ensures stable predictions that are independent from the defined parameter. For the topic of content generation, *Mastropaolo et al. (2021)* pre-train a T5 model on a data set composed of natural language English text and source code. The model is fine-tuned by reusing data sets to improve code content such as automatic bug-fixing, injecting code mutants, generating assert statements in test methods, and code summarization. For the automatic bug-fixing task, the model's performance is similar to the baseline results, the same for generating assert statements in test methods and code summarization. In case of injection of code mutants, the model performs better than the baseline with an increase in accuracy of 11% (*Mastropaolo et al., 2021*). However, the validity of the findings is open to discussion. The study raises concerns regarding the data set splitting for pre-training and testing: a code comment among the pre-training instances can have a duplicate in the test set of the code summarization task. The strict separation between training and testing data sets is crucial to ensuring the validity of the model performance. An interesting aspect of this study is the characteristics of the T5 model. Given that the experiments were conducted using six data sets, the T5 model is language agnostic, facilitating its application in different programming languages. For the text summarization area, the article (*Gavrilov, Kalaidin & Malykh, 2019*) presents a new approach to headline generation based on the Universal Transformer architecture. The architecture presented manages to explicitly learn non-local representations of the text. The proposed transformer architecture learns non-local dependencies between tokens regardless of the distance between them. This approach allows the model to assimilate a more complex representation of the text and demonstrates that this is a mandatory strategy for the text summarization problem. The headline generation performance is tested on two data sets, the New York Times, and the Russian news agency. The model proposed in this study has limitations because it does not achieve human similarity, leaving space for further improvements.

## DISCUSSION

Based on the above analysis, there are six main applications that utilize transformer-based approaches, each of them presenting particularities. The applications are presented in Table 1. In our review, we have identified that the most common topic is text classification, and the most popular architecture is BERT (Table 3). In text classification studies using BERT, we have identified limitations such as inefficient use of input data and BERT performance comparable to traditional approaches (*Rasmy et al., 2021*; *Balagopalan et al., 2020*). One possible solution could involve exploring hybrid methods that combine both classical ML solutions and transformers, and refining feature definitions to fully leverage the information provided by input data. We have identified limitations in BERT approaches for NER and sentiment analysis applications. For example, within the biomedical domain, a study focused on the extraction of clinical concepts uses BERT for a word-level extraction task (*Yang et al., 2020*). Meanwhile, BERT has shown promising performance in sentence-level applications such as classification (*Rasmy et al., 2021*) or text summarization (*Yang et al., 2020*). For sentiment analysis tasks using BERT, the study on hate speech detection on online social media (*Mozafari, Farahbakhsh & Crespi, 2020*) encounters errors due to annotation rules or biases in the input data. Another study for the detection of hate speech in different languages (*Sohn & Lee, 2019*) is devoted to the ability of Transformer to understand multiple characteristics of the language such as irony or sarcasm. Possible solutions for these problems can be an in-depth analysis of contextual data to identify features that can be used to fine-tune BERT parameters or identify text data augmentation methods to introduce additional value to the input text and enrich the complexity of the initial data set. Furthermore, for the optimization topic, we have identified innovative NLP studies that utilize Transformers for quantization tasks, while preserving model performance (*Zafrir et al., 2019*). The study focusing on BERT quantization enables the integration of NLP applications on edge devices, demonstrating promising results with minimal accuracy loss and reduced latency.

Finally, GPT architectures are becoming very popular considering the performance demonstrated in content generation tasks and their potential to address NLP applications (Tables 1 and 3). In our review, we have identified an innovative solution that aims to connect information from two different data sources: text and video. The study has shown that GPT can effectively leverage information presented at the text level and video level and capture interaction between different sources (*Li et al., 2021*). Another study uses GPT-2 to develop an agent that generates empathic responses in online mental health support (*Sharma et al., 2022*). However, the sensitive nature of this application still requires human involvement for evaluation.

Like most studies, the design of the present study is subject to limitations. We have chosen the review methodology provided by *Petersen, Vakkalanka & Kuzniarz (2015)* because it provides strategic guidelines for conducting a thorough review. The inclusion and exclusion criteria are included in Petersen's methodology as a fundamental principle. For this review, we adapt this principle to suit our specific objectives. Consequently, we employed a series of filtering methods with a focus on the most important NLP studies. We

are confident in the methodology provided by *Petersen, Vakkalanka & Kuzniarz (2015)* and aware of the selection limitations in our current approach. First, the selection and comprehensiveness of the articles included in the review can be influenced by the choice of keywords and search terms. Second, the Total Smart Citation score used as an inclusion criterion can be calculated only on articles that contain the DOI number. While this research may result in incomplete coverage of the literature and potential omission of relevant studies, it is important to emphasize that our objective was to focus on studies with high impact in the NLP research area with Transformers. In future research, we aim to overcome the above-mentioned implications by adopting a less rigorous selection process. However, we are committed to ensuring the quality of the articles included in the review. We are confident that the current selection process, incorporating Total Smart Citation, allowed us to focus on articles with significant impact within the NLP research domain.

## Future directions and challenges

NLP has experienced significant improvements along with the emergence of Transformers and large language models (LLMs). The models allow different approaches to various NLP tasks and enable addressing a wide variety of domains. For example, within the medical domain, *Rasmy et al. (2021)* successfully tackled the challenge of disease prediction from electronic health records, while *Lee et al. (2020)* and *Yang et al. (2020)* effectively addressed complex NLP challenges like biomedical text mining and clinical concept extraction. Significant progress has been demonstrated in other areas of NLP, like language-specific tasks. For instance, *Farahani et al. (2021)* developed a Transformer-based model for Persian language understanding. Additionally, we have identified other language-specific tasks, such as hate speech detection (*Sohn & Lee, 2019*) and false information prevention (*Ayoub, Yang & Zhou, 2021*).

As for future research objectives, one potential direction could include multimodal approaches for enhancing NLP tasks by combining different types of information. One of the challenges of this particular task is the management process for different data types. Thorough preprocessing methods are required to ensure that the most important data characteristics are identified and fed to the NLP model, thereby improving the results. Another challenge in the NLP field is the necessity for data set enhancement. A possible solution for this problem is text augmentation techniques that can be used to increase the size of the initial data set. The data set enhancement task can be tackled by applying deep learning methods to create new features and enrich the complexity of the existing text data. Hybrid approaches, combining transformers with traditional machine learning methods, represent another direction for future research that can improve the outcome of NLP tasks. In the text classification problem, hybrid approaches can achieve better generalization within classes, leading to overall optimized results. Finally, a future direction worth exploring involves the challenges and opportunities within domain fine-tuning. Good results that outperformed traditional methods were acquired by model fine-tuning in various tasks (*Yang et al., 2020*; *Mozafari, Farahbakhsh & Crespi, 2020*; *Sohn & Lee, 2019*; *Balagopalan et al., 2020*). However, the LLMs eliminate the necessity of fine-tuning in NLP

tasks. This advantage elevates the new LLMs above previous state-of-the-art models and paves the way for further experimentation across the NLP domain.

## CONCLUSIONS

In recent years, NLP approaches have proven to be increasingly efficient in solving various human language tasks. BERT-based models, GPT architectures, and other Transformers successfully overcame problems from different areas of interest. This study reviewed NLP solutions with Transformers that could be categorized into six applications, the most common being the text classification domain. Additionally, by analyzing the data sets, we identified and classified the studies into four distinct types, providing a systematic classification based on the characteristics of the data sets. The challenges and limitations that occur in NLP applications are closely dependent on the Transformer architectures. Therefore, this review presents some of the research gaps from an architecture perspective. We expect that efficient transformer training and a thorough study of the possibilities offered by NLP methods can overcome the language limitations emphasized in this study. By conducting this study, we identified compression approaches for BERT models that succeeded in reducing the memory footprint without decreasing the performance for NLP tasks. This represents an essential outcome for the successful use of edge architectures in the NLP field. Furthermore, GPT and its derivative architectures have demonstrated promising performance in text generation tasks, facilitating the development of automated NLP solutions. These findings could serve as a valuable contribution for linguistics specialists and computer science developers with a shared interest in NLP with Transformers.

### Funding
The authors received no funding for this work.

### Competing Interests
The authors declare that they have no competing interests.

### Author Contributions
- Georgiana Tucudean conceived and designed the experiments, performed the experiments, analyzed the data, prepared figures and/or tables, authored or reviewed drafts of the article, and approved the final draft.
- Marian Bucos conceived and designed the experiments, performed the experiments, analyzed the data, prepared figures and/or tables, authored or reviewed drafts of the article, and approved the final draft.
- Bogdan Dragulescu conceived and designed the experiments, performed the experiments, analyzed the data, prepared figures and/or tables, authored or reviewed drafts of the article, and approved the final draft.
- Catalin Daniel Caleanu conceived and designed the experiments, performed the experiments, authored or reviewed drafts of the article, and approved the final draft.

## Data Availability

This is a literature review.

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
