# Peer review of "Natural language processing with transformers: a review"

_PeerJ Computer Science, doi:10.7717/peerj-cs.2222_

## Round 0.1 · original submission · Major Revisions

The contribution is interesting and a number of reviewers think they can be useful for the community with some changes. Therefore, we encourage the authors to address deeply the suggestions and try to improve the article.

**Language Note:** PeerJ staff have identified that the English language needs to be improved. When you prepare your next revision, please either (i) have a colleague who is proficient in English and familiar with the subject matter review your manuscript, or (ii) contact a professional editing service to review your manuscript. PeerJ can provide language editing services - you can contact us at [email protected] for pricing (be sure to provide your manuscript number and title). – PeerJ Staff

Reviewer 1 ·

Basic reporting

The review is written in a correct English which ease the labor of reading and understanding. In addition, the sections that are required for a review are successfully fulfilled. Just one point to remark here:

1 . The reference number [23], is not correct because it does not reference to a paper, instead references to the plugin of Zotero. Keep in mind that [23] is used in the review as complementary information to support the main idea that is exposed in the document. So, correct that reference.

Experimental design

I appreciate the deep research done by the authors and the well structures and explained that is the information. Nevertheless, there are some points that I would highlight and which the authors should face to solve them:

1. In lines [65-74], you explain the basic idea of GPT but you just underscore one or two differences among their versions. You must explain more deeply about what is or how works GPT and which are the main differences between BERT models and GPT because the differences are not clearly exposed. Moreover, you must highlight more differences among the versions of GPT due to the lack of explanations that are given in the document.

2. In line 70, explain what do you mean by autoregressive model?

3. In line 367, you wrote that “These limitations could be mitigated in future research employing alternative methodologies.”. As I see also in Line 124, you used the method proposed by Petersen et al [21]. So, considering these two points, explain why you concretely chose this methodology among the others that exist and named the “alternative methodologies" that you are referring to.

Validity of the findings

The references used give a great support to the information that you are exposing. I would like to mention one aspect that would hone the document and the authors must consider:

1. You did a great job using the references to specify the paper where Transformer Models (T5, Universal Transformer, X-Transformer) are used so the lector can follow the references in case he wants to look for more additional information. However, this is not the case when you write about the classification of datasets (specialized, publicly, benchmark, created). I considered that would be helpful if you named some datasets of each type and their respective references where the lector can find them.

Additional comments

The literature review carried out by the authors is a great scientific contribution that can be useful to many researchers. I consider that this review has been carried out thoroughly in the sense that an orderly and clearly specified methodology has been used to carry it out. In addition, the review has considered taking articles less than 5 years old, which allows taking into account the latest scientific contributions, which are the most updated. This fact is very important in the field of AI where everything evolves rapidly. However, the previous points must be covered successfully by the authors in order to make better the review.

Cite this review as

Reviewer 2 ·

Basic reporting

The paper presents a review of studies that use transformer-based models in NLP. The study does not provide any particularly new or valuable information to the audience in the field. The paper lacks a clear
and well-defined research objectives and does not thoroughly analyze the unexplored research gaps that motivated such a study.

Experimental design

a very large number of studies have utilized transformer-based models for NLP problems, however, due to the selection criteria (Total Smart Citations, Non-BibTeX Studies, etc.) only 128 studies were considered in this research. This might lead to the exclusion of some important and promising studies that might have valuable insights or innovations.

Validity of the findings

the paper lacks an in-depth analysis of the relevant studies. It mainly focuses on counting the number of studies related to specific NLP tasks but does not provide a comprehensive examination of these studies, their methodologies, key findings, or their impact on the NLP field.
I recommend that the authors consider incorporating a more detailed analysis of the selected studies to provide a richer understanding of the research field.

Additional comments

no other comment

Cite this review as

Reviewer 3 ·

Basic reporting

The authors in this paper titled "Natural language processing with transformers: A review" aimed to to highlight future research directions that include NLP applications with Transformers. The manuscript needed to be improved from various aspects as different sections of the paper are not well aligned.
The title is not reflecting the work done in this study, Abstract gives a different aim of the paper which has is not seen being achieved in the paper. The motivation and research contributions being made are not clear as well.

Experimental design

In the selection of articles no years/period is mentioned. Moreover, the reach question are not defined properly and how the research questions are being answered by the work done in this paper.

Validity of the findings

The aims set in the abstract to highlight future research directions that include NLP applications with Transformers is not achieved by the discussion presented. A more detailed and structured approach is required.

Cite this review as

·

Basic reporting

According to the authors, the study has conducted a step-by-step process: identify the recent studies that include Transformers, apply filters to extract the most consistent studies, identify and define inclusion and exclusion criteria, assess the strategy proposed in each study, and finally discuss the methods and architectures presented in the resulting articles.

Experimental design

The authors have nicely classified the articles according to different NLP problems (mainly in 6 types) in the “Result” section. The authors have discussed some technical details in the “Discussion” section.

Validity of the findings

However, the authors have mentioned the methods in an informative way; from their “Result” section, it is difficult for the reader to find the technical details of the methods and models used for the particular NLP problem. But, the “Discussion” section is not organized according to different types of NLP problems, which can confuse the reader. From conclusion statement - "By conducting this study we identified compression approaches for BERT models that succeeded to reduce memory footprint without decreasing the accuracy and enabling low-latency for NLP applications. This represents an essential outcome for successful usage of EdgeAI architectures in the NLP field. Furthermore, GPT and its derivative architectures continuously demonstrate outstanding performance, so studying and applying similar architectures can improve the potential of the NLP tasks that we aim to address in future research.", the authors have remarked in the "Abstract" that -"We have identified some of the newest trends in the NLP Transformer field – Generative Pre-Training (GPT) architectures and Bidirectional Encoder Representations from Transformers (BERT).". But I did not find anything in their paper that supports their last statement in the "Abstract" - "The final goal of this review is to highlight future research directions that include NLP applications with Transformers."

Additional comments

no comment

Cite this review as

---

## Round 0.2 · Minor Revisions

Dear authors,

After considering the reviewers' comments, I recommend to revise again the manuscript. Most of the comments are addressed by authors, however, one of the comments is not satisfactorily addressed according to one of the reviewers. Please, it is important to include a detailed section on open research issues and future directions to strengthen research contributions made by this paper.

Reviewer 1 ·

Basic reporting

As it was highlighted in the first review done, the reference [23] is not correct because it refers to Zotero's plugin. Correct that reference and make sure that it refers to the correct paper.

Experimental design

All the suggestions that were given in the 'Study Design' section were successfully fulfilled.

Validity of the findings

As in the previous section, the authors took into account the requested considerations.

Additional comments

The authors have completed most of the points that were requested. They did a great job but they must review the reference [23] in order to finish the corresponding corrections.

Cite this review as

Reviewer 3 ·

Basic reporting

no comment

Experimental design

no comment

Validity of the findings

Thank you for making efforts to address the raised comments. However, I believe one of the comments can be addressed in a better way that might require some time and effort but will eventually improve the quality and research contributions.

Please refer to this previous comment.
"The aims set in the abstract to highlight future research directions that include NLP applications with Transformers is not achieved by the discussion presented. A more detailed and structured approach is required."

It is suggested to add a detailed section on the open research challenges and future directions for NLP with transformers.

It is also important to briefly cite and acknowledge the power of LLMs for NLP tasks.

Cite this review as

---

## Round 0.3 · accepted · Accept

I confirm that the authors have addressed all of the reviewers' comments successfully, so the manuscript is ready for publication.

Reviewer 1 ·

Basic reporting

The point that was requested in the previous review has been successfully fulfilled.

Experimental design

No comment.

Validity of the findings

No comment.

Cite this review as